# A Survey on the Application of WirelessHART for Industrial Process Monitoring and Control

**DOI:** 10.3390/s21154951

**Published:** 2021-07-21

**Authors:** P. Arun Mozhi Devan, Fawnizu Azmadi Hussin, Rosdiazli Ibrahim, Kishore Bingi, Farooq Ahmad Khanday

**Affiliations:** 1Department of Electrical and Electronics Engineering, Universiti Teknologi PETRONAS, Seri Iskandar 32610, Malaysia; arun_18003272@utp.edu.my (P.A.M.D.); rosdiazli@utp.edu.my (R.I.); 2School of Electrical Engineering, Vellore Institute of Technology, Vellore 632014, India; kishore.bingi@vit.ac.in; 3Department of Electronics and Instrumentation Technology, University of Kashmir, Srinagar 190006, India; farooqkhanday@kashmiruniversity.ac.in

**Keywords:** automation, control system, fractional-order control, industrial wireless sensor networks, network control system, process control, WirelessHART, wireless control

## Abstract

Industrialization has led to a huge demand for a network control system to monitor and control multi-loop processes with high effectiveness. Due to these advancements, new industrial wireless sensor network (IWSN) standards such as ZigBee, WirelessHART, ISA 100.11a wireless, and Wireless network for Industrial Automation-Process Automation (WIA-PA) have begun to emerge based on their wired conventional structure with additional developments. This advancement improved flexibility, scalability, needed fewer cables, reduced the network installation and commissioning time, increased productivity, and reduced maintenance costs compared to wired networks. On the other hand, using IWSNs for process control comes with the critical challenge of handling stochastic network delays, packet drop, and external noises which are capable of degrading the controller performance. Thus, this paper presents a detailed study focusing only on the adoption of WirelessHART in simulations and real-time applications for industrial process monitoring and control with its crucial challenges and design requirements.

## 1. Introduction

The network control system has been widely adopted in process industries and manufacturing plants producing goods such as food and beverages, chemicals, pulp and paper, crude oil refineries, and power generation plants to control and monitor field instruments. These industrial control systems consist of a single instrument or a group of instruments that form a single- or multiple-loop network based on their design and deployment. To maintain a steady output response in a system, the industrial control system must continuously monitor and maintain the parameters at desired levels. Hence, industries need to control and monitor numerous field devices that humans often handle. In the initial industrial era, communication between field devices and control systems took place via 4–20 mA analog signals, causing more errors that resulted in process instability and additional external noise [1,2]. Thus, the concept of applying automated networking control systems to industrial processes has grown in many applications, yielding autonomous controllers without the need for any human intervention. This has led to the hybridization of combining analog and digital signals to create new wired communication protocols. These include, to list a few, FOUNDATION^TM^ Fieldbus, Modbus, PROFIBUS, ISA 100.11a, and highway addressable remote transducer (HART), as well as their upgraded wireless versions, such as ZigBee, WirelessHART, ISA 100.11a wireless, and WIA-PA [3,4]. For an industrial wireless sensor network (IWSNs) to be used in industry, it must provide the same quality of controlling and monitoring service as—or better than—conventional wired communication systems [5,6]. Some of the advantages of IWSNs over wired networks are:They can eliminate the costly and bulky cabling used to connect the various field devices [7];They will dramatically reduce the deployment, redeployment, installation, and commissioning times, thus avoiding the problem of frequent cable maintenance [8];They will be self-organized and support a large number of battery-powered wireless nodes [9];They can be installed at any location irrespective of the surrounding environmental conditions [10].

The advancement in communication technologies changed control strategies from being based on electronic single control loops in the 1960s to single digital loop controllers in the 1970s. Multi-loop digital controllers were designed for single process plants in the 1980s. The present wireless digital controller is shown in Figure 1 [11]. Self-diagnostic functionalities in wireless field instruments have encouraged process industries to shift from wired to wireless technologies, supporting both analog and digital instrument installation in any environment. Figure 2 shows a classical IWSN architecture, where multiple wireless nodes are placed in plants remotely and controlled through wireless networks. The closed-loop control system consists of both continuous-time and discrete-time data of the sensors, actuators, and controllers integrated with the process plant. In the closed-loop system, both the host application and the gateway will run on the same system to avoid the possible delay and clock drift. This closed-loop structure is the same in all existing wireless networks.

The overall classification of the IWSN communication protocols is shown in Figure 3. From the figure, it can be seen that all the IWSN standards are based on the physical layer IEEE802.14.5 standard. However, modifications were carried out in the physical layer to suit the specific application needs of each standard. A significant characteristic of the WirelessHART is its time-synchronized MAC layer specification [4]. Additionally, its MAC header is designed in such a way that it supports the co-existence of other IWSN protocols, such as Wi-Fi, ZigBee, ISA 100.11a, and WIA-PA. Table 1 shows a summary of a survey conducted by ISA, HART Communication Foundation, and Wireless Industrial Networking Alliance in collaboration with “ON World” to understand the factors influencing the adoption of IWSN among vendors and end-users [12,13]. The foremost concern in the adoption of industrial applications is data accuracy and data protection against malware attacks and hacking. The next concern relates to easy data access for field devices and the adaptability of universal industrial standards. In addition, industries are unwilling to take the risk to deploy large-scale sensor nodes unless widely accepted industrial standards back them. Due to these reasons, industries are seeking inter-operable standards to deploy wireless field devices. Additionally, the crucial research improvements in IWSNs increased the market cap value from $944.92 million to $3.795 billion [3]. In addition, this trend is continuing to grow even faster in the forthcoming years due to the rapid industrial developments (Fourth Industrial Revolution (IR 4.0)). To deal with the anticipated trends, more research development solutions focusing on signal reliability, inter-operability, compactness, effective data transmission, and fault tolerance characteristics are essentially needed for the IWSN protocols.

Table 2 summarizes the comparison between features of the most commonly used industrial wireless standards. In the table, it can be seen that the three industrial standards WIA-PA, ISA100.11a, and WirelessHART share several features in common. These features include security, reliability, scalability, topology, and low power consumption. The main objective of the paper is to survey the implementation of WirelessHART in industrial process control in both simulations and real-time environments. Figure 4 gives the hierarchical flow of the paper organization. The figure gives brief information about every section present in the article. Additionally, other important contributions to the field of process control from this research paper will be given as follows:Evolution of the IWSN with its architecture and classification;Progression of the industrial process automation using IWSN;How the WirelessHART protocol dominates the process control industries;WirelessHART network architecture with its OSI layer structure;Detailed survey of the utilization of WirelessHART for industrial process control in simulation and real-time implementation;Design challenges and application-based requirements for the WirelessHART network;Possible research and development solutions for the WirelessHART network requirements and challenges.

Furthermore, the scope of the survey presented in this paper is different from that of other review articles. Numerous interrelated research articles were utilized to provide an extensive literature analysis of the WirelessHART implementation in industrial process control. Here, all of them are organized based on several factors, such as network analysis, type of field device used, network topology, simulator tool used, and the controller used in both the simulation and real-time environments with its challenges and design requirements. Comprehensive studies of the existing IWSN communication protocol’s network architecture, design, and standardization can be found in [3,6,15,16,17].

This paper’s remaining sections are organized as follows: Section 2 provides an introduction relating the importance of wireless networks in process control automation with the classification of different industrial processes with their functions. Section 3 gives a detailed analysis of WirelessHART in process monitoring and control applications in simulations and real-time environments, along with its industrial network architecture. Section 4 describes the challenges and design requirements for WirelessHART with various limitations and its possible research solutions, followed by a summary and conclusion in Section 5.

## 2. Background of Process Automation

For many decades, industrial process plants have used analog signals in the wired channel for communication with field devices (sensors) to take appropriate control actions to ensure process stability [15,18]. The most commonly used communication protocols, such as HART, Fieldbus, Modbus, and PROFIBUS, emerged in the mid-1980s. However, wireless technological advancements required the development of industry-standard wireless protocols for them to be used in process monitoring and control. These wireless communication protocols require small- to mid-scale network infrastructures consisting of multiple sensor nodes working together to acquire data from field devices installed in different environments. Their design is based on the application-specific requirements, since each industrial process has multiple objectives and different infrastructure needs [3]. In the past few years, IWSNs have emerged in numerous application fields, including personal health monitoring [19], building and civil infrastructure monitoring [20,21], automotive applications [22], power converters [23], power and smart grids [24], energy harvesting [25], smart cities [26], agriculture [27], food processing [28], underwater wireless sensor networks [29,30], and environment monitoring [31].

Process and industrial automation fields, such as steel manufacturing, oil and gas, pulp and paper, and power generation, have started to gradually adopt IWSNs because of the new technological advancements made and the possible flexibility in handling complex closed-loop processes [32]. In industrial processes, they were expected to achieve about 80% of the market share in 2020 by overtaking wired networks at the field level due to their efficient and easily deployable infrastructure [33]. The main reason for adopting wireless motes is due to their operational and installation cost reduction of up to 60% in comparison with conventional wired field devices, according to an industry operation survey overseen by Emerson Process Management [34,35].

### 2.1. Process Control Automation

The applications of process control can be classified into three distinct sub-categories based on the control system point of view, as presented in Table 3 [36]. A brief classification for each is given underneath.

#### 2.1.1. Safety and Supervisory Control

The transmission of sensor data to the controller in safety and supervisory control is very much essential. Additionally, issues such as packet loss and latency cannot be tolerated because these are emergency control systems, and their failure will lead to catastrophic accidents. Thus, sensors connected to these systems are always in standby mode, with a maximum permitted latency of 10 ms.

#### 2.1.2. Closed-Loop Control

Closed-loop control is a conventional system that has a controller maintaining the desired set-point of the process. Here, dead-time and external noise cause significant issues, while the maximum allowed latency varies from 10 to 100 ms, with a less critical rate in comparison with emergency class systems.

#### 2.1.3. Monitoring and Control

In this classification of control systems, latency is not considered an essential factor in taking control actions and there is a maximum allowed latency of 1000 ms. Here, field device data are commonly utilized to perform maintenance operations for calibration and repair. However, in some cases, data transfer consistency is needed to continue the process operations.

### 2.2. Evolution of Wireless Networks in Process Automation

Initially, the ZigBee wireless standard was developed to monitor and control different home automation products. Later, it was extended for specific industrial processes, but it was not suitable for regulatory and emergency classes because of its poor data reliability. ZigBee is highly suitable in monitoring and alerting systems, where energy savings is given priority [36]. The remaining communication protocols were explicitly developed for factory automation applications, where each of them was designated for various industrial application classes. WirelessHART, for example, was designed to support closed-loop supervisory and regulatory applications because of its efficient routing capabilities and high potential communication between multiple field devices to ensure multi-channel frequency hopping [37,38]. ISA100.11a and WIA-PA are intended to provide more flexible coverage over all classes of industrial processes listed in Table 3. All these protocols use IEEE 802.15.4 as a physical standard and have a MAC layer with an equal number of channels.

On the other hand, emergency systems require a latency of not more than 10 ms, reliable data transmission, and mote parity. Thus, for these kinds of systems using a multi-hop network is not a suitable option because of network stability issues [6]. The communication standards examined here were mainly developed for monitoring and control category applications, such as open-loop and alerting systems, as shown in Table 3. The preferred standards among the existing IWSNs in industries were surveyed in 2012 and 2014; the results are summarized in Table 4 [12]. The results indicated that one out of four users preferred WirelessHART, even though it faced a slight decline in the number of adopters. ISA100.11a has attracted adopters, which has resulted in a marginal growth in its implementation. The remaining wireless standards adopted among industrial users are WIA-PA, ZigBee, and Factory Automation. This gives WirelessHART a clear lead for use in the process automation industry [39,40].

## 3. Industrial Applications of WirelessHART

This section gives a brief introduction to the WirelessHART protocol, its typical network structure, and different OSI layers usage. Furthermore, a detailed review of the application of WirelessHART for industrial process monitoring and control in both simulation and real-time environments will be discussed.

### 3.1. WirelessHART

The evolution of the HART protocol is shown diagrammatically in Figure 5. From the figure, it can be seen that since 1988, with only around 4 million wired devices, the standard has incorporated devices such as digital control valves and controllers with HART6 by 2002. By 2007, EDDL and wireless technology were integrated into the latest version of the HART protocol (HART7), released as WirelessHART (IEC 62591), which is the first wireless communication protocol to adopt an over 2.4 GHz radio frequency channel in the IEEE 802.15.4 for industrial process control applications [41].

WirelessHART, being based on the traditional HART protocol, has already gained wide patronage in the industry due to the necessity of demand in the open international standard that suits industrial requirements. The latest version (version 2) of the WirelessHART protocol was approved by the International Electrotechnical Commission in 2016. The standard possesses some new updated features, such as:Wireless mesh networking;Time synchronization and stamping;Network and transport layer;Security encryption and decryption;Enhanced burst mode messaging;Pipes for high-speed file transfer.

The WirelessHART communication protocol utilizes only five layers of the OSI model out of the seven layers. Figure 6 shows the usage of different OSI layers between the conventional wired HART and the WirelessHART protocols. The five OSI layers used by WirelessHART are the physical layer, the data link layer, the network layer, the transport layer, and the application layer. Routing, communication scheduling, and corresponding signal generation are handled by the central network manager. Further detailed discussion of the various OSI layers of WirelessHART and other IWSN communication protocols can be found in [6,42].

An added advantage of WirelessHART is that it can be extended to control the process rather than simply monitor it. Wireless local area network (WLAN), Bluetooth, ZigBee, and Internet Protocol Version 6 (IPV6) are not extensively adopted for industrial wireless applications because of their limitations in controlling capabilities. At present, two of the most widely used industrial international wireless standards are WirelessHART and ISA100 Wireless [44,45]. Among these two, WirelessHART leads with more than 30 million installed field devices, and it is projected that this figure will reach over 46 million by 2021 [46]. Hence, there will be very little or no need for training the plant operators to start using the WirelessHART. Based on its flexibility, interoperability, simplicity, and acceptability, the WirelessHART has many advantageous over the ISA 100.11a standard. Simultaneously, the ISA 100.11a standard is yet to gain approval from the International Electrotechnical Commission (IEC). This has given WirelessHART supremacy in industry [47]. Both standards aim at non-critical wireless applications for control and monitoring purposes. Nevertheless, WirelessHART is generally preferred by industries since its legacy wired HART communication protocol was once the dominant and most widely adopted protocol in industrial field devices. Additionally, converting existing wired HART field devices to wireless ones is less costly and there is no need for additional sensor components [48].

The WirelessHART network control system (WHNCS) structure is shown in Figure 7. There are five essential elements present in the WHNCS, namely:Field device: connected to the industrial process plant.Wireless handheld: employed for diagnostics, device configuration, and calibration from a remote location.Gateway: acts as a bridging device to connect host applications and field devices.Network manager: accountable for configuring the network, scheduling, routing, and managing communication.Security manager: managing and allocating security encryption keys and keeping track of authorized devices to connect to the network.

### 3.2. Simulation Environment

After the emergence of WirelessHART as the first wireless standard for monitoring and controlling industrial applications, attempts were made to evaluate them in a simulation environment using network simulators, hardware-in-the-loop simulator combined with Matlab, and LABVIEW. Jouni et al. [49] first acquired the patent for WirelessHART communication to control a field device in an industrial process using a cellular communication system. Here, the control system receives data from an internet-connected field device with a diagnostic system connected to it. This method of transmission increases the time delay and is prone to security threats. Later, improvements to security and co-existence with IEEE 802.11g networks were proposed in [50]. Their results validated the network’s adaptive frequency hopping ability, rejected high packet loss channels, and blacklisted them to improve reliable data transfer. They concluded that security measures still need to be improved to counter denial-of-service attacks.

Numerous researchers carried out various analyses of protocol development, performance, interoperability, and simulation investigations for process control during the WirelessHART establishment period to understand its effectiveness [51,52,53]. The initial attempt to use WirelessHART for control-oriented processes started with the development of TrueTime, a Matlab/Simulink-based wireless simulation toolbox specifically designed to support WirelessHART [54]. This modified TrueTime toolbox was used by researchers to simulate a closed-loop process control system in the WirelessHART standard with various packet losses, clock drifts, and delay compensation conditions [55,56,57]. Communication scheduling and controller design methodologies were combined to form a co-design technique to minimize control systems communication latency when using the WirelessHART standard. This method addresses the real-time issues in end-to-end data reliability, packet scheduling, packet loss/drop, and controller performance [58]. A WirelessHART-based simulator focusing on industrial process control application is presented in [59]. This simulator utilizes all 15 channels available in the communication standard for effective scheduling and data transfer to avoid network interference by using multi-hop communication.

WirelessHART’s network performance was evaluated against a hybrid simulation approach using COOJA in the Contiki operating system, with particular attention given to its efficient memory and time slotting. Though this method supports the handling of multiple industrial systems, it only supports one data argument. Other essential parameters, such as network management, communication scheduling, flash memory usage, and WirelessHART compliant security layer implementation have not been adequately addressed [60]. An improved co-design simulation technique using an interference model of the process was coded in OMNET++, which is used in TrueTime-Matlab/Simulink [61]. The simulation was conducted for monitoring and controlling the DC servomotor over a WirelessHART network with a conventional PI controller in a closed-loop process. In this model, prominent factors such as multipath fading, noise from the environment, signal interference, and line of sight are not considered, which makes the network reliability questionable. In [62], a hybrid control-oriented approach using WirelessHART in NCS with a source routing configuration to achieve asymptotic and exponential stability under some constraints is presented. In this research, important communication constraints such as stochastic time delay, interference, and packet drop are not examined.

In [11], a simulation using Fast Sampling Wired Link Contention in a WirelessHART network control system with conventional PID is presented to improve the link reliability. Link delay and packet dropout correlation factors are used to address its impact on the closed-loop control performance. They improved the system efficiency by adopting an exponentially weighted moving average (EWMA) filter to remove the packet collisions. In [63], a formation of a distributed WirelessHART network is created by adopting field-level scheduling through a time window slotted allocation. This properly scheduled transmission reduced the power consumption up to 85% and enhanced the network scalability in comparison with the centralized method. Furthermore, an additional detailed discussion of the WirelessHART simulation by various researchers with their controllers, field devices (virtual nodes in the case of the simulation), and network structures is presented in Table 5.

### 3.3. Real-Time Implementation

Song et al. [64] initiated research on applying WirelessHART in real-time industrial process control and demonstrated their results. The researchers used a modified Freescale 1321xEvk toolkit written in the ANSI C language and created a super-frame time slot configuration for the hosted devices. Simple data scheduling and transmission between the field devices were carried out to indicate the possibility of monitoring and data transfer using the WirelessHART Network. In [65], multiple control strategies for WirelessHART network devices are proposed—namely, (1) controlling through the host, which supports complete control; (2) controlling through the field, which supports partial control; and (3) controlling through the gateway, which supports full control and has less latency compared to all the other approaches. They used the WirelessHART temperature transmitter to transmit and acknowledge the data transmission through the gateway without controlling the process. Real-time experimentation on the distillation column pressure and steam flow was conducted to prove that control over WirelessHART is possible [66]. The process was controlled using a conventional PID controller in both cases to maintain the process set-point. The results proved that WirelessHART transmitters improved the accuracy and performed as reliably as the wired communication without signal filters. Additionally, wireless transmitters are not affected by ground loops, which often affects wired field devices. LabVIEW-based WirelessHART experimentation was conducted to study the effect of packet loss on the network control system stability [67]. Multiple industrial communication protocols were compared regarding the gradual increase in the loss probability of data packets from 0 to 100% to understand their impacts on a network control system’s stability.

The PIDPlus algorithm, an improved variant of the conventional PID controller, was employed to take care of slow process updates, packet loss in the communication channel, and non-periodic measurement updates encountered by WirelessHART transmitters [68]. Two other studies using a PID controller with a Kalman filter and Smith predictor were conducted to compare it with the PIDPlus algorithm. The PID with the Kalman filter had a better performance in terms of integral absolute error (IAE). Developments for monitoring and controlling the dividing wall column using event-based model predictive control (MPC) were carried out in [69]. MPC outperformed the PID controller, with faster handling and better compensation for the external disturbances. It suffered from network-induced delay and the implementation complexity increased due to the greater number of tunable parameters. Experimentation with an internal model control (IMC) aimed to investigate the disturbance rejection and set-point tracking capabilities in an industrial-scale pilot process plant. IMC showed a better performance in set-point tracking and overshoot reduction than the PID controller, but when the system order increased the IMC had a slower rise time and increased peak overshoot, which caused the system to settle as equal PID. In [14], the Smith predictor-based filtered predictive PI (FPPI) controller was analyzed in an attempt to compensate for the model mismatch and high-frequency noise issues faced by IMC. Additionally, this controller possessed useful time-delay prediction capabilities to handle stochastic systems. This controller has a simple design structure and has the same number of tunable parameters as the conventional PID controller, which led to its easy implementation and robust performance. Figure 8 shows a comparison of the IAE values for different controllers in the WirelessHART network for various industrial processes. Meanwhile, the tunable parameters possessed by the different controllers discussed above are shown in Table 6 [70].

The parameters discussed above are some of the major contributions towards the real-time implementation of WirelessHART in process control. A detailed study of the real-time implementation of WirelessHART in different controllers, industrial applications, topologies, software tools, and other parameters is given in Table 7. The majority of the papers concentrated primarily on overcoming the problems of latency and data reliability.

## 4. Challenges and Design Requirements

The deployment of wireless motes comes with its own requirements and challenges because of the significant differences between the office environment and the industrial environment. For example, in the industrial environment, the deployed wireless network must have the ability to support the low latency with secure data communication in critical processes. For inherent safety, these networks must possess a high fault-tolerance capability and have highly reliable data transmission in order to meet the industrial requirements [92,93]. Additionally, low-power wireless motes need to be developed, since they have to operate for more extended periods between major turnarounds considering the harsh conditions [94]. In the future, the implementation of non-conventional energy-powered mote installation is anticipated to increase in industrial and process automation because it dramatically reduces power blackouts, avoids the problem of battery replacement, and has a smaller carbon footprint [27].

The deployed WirelessHART network should be aware of unpredictable environmental parameters such as temperature, moisture, gas level, pressure, and vibrations in the environment where the field devices are located. Wireless mote signals could be severely delayed due to various circumstances, such as interference with multiple frequency bands, reflections from the surrounding walls, external noise, signal attenuation due to leaked gases, and vibrations produced by heavy machinery [95,96]. Most of the problems stated above will affect their deployment because a minor interruption can make the network less reliable and may lead to catastrophic failure in the process. Other major influencing parameters in the WirelessHART network are briefly discussed in the following.

### 4.1. Security

Security is one of the prime challenges in WHNCS deployment in the process control industry. Irrespective of critical and non-critical processes, they are always prone to security threats [97]. To achieve secure transmission, the network should be aware of denial-of-service (DoS) attacks and cyber-attacks from outside networks [98]. Both active and passive attacks may take place, such as snooping on transmission signals, the modification of signal information, signal interruption, data flooding, and re-routing the network paths. These must all be considered in the design phase [3,99]. Due to resource limitations, security protocols have to be balanced against other quality of service (QoS) performance requirements [100]. Data encryption, data authentication, and cluster-based private data aggregation (CPDA) techniques will minimize security attacks in the network. Sensor network encryption protocol (SNEP), localized encryption and authentication protocol (LEAP), and random key pre-distribution (RKP) are some of the most effective techniques to block malicious attacks such as data flooding, information spoofing, and data transit attacks [101].

### 4.2. Reliability and Interference

In the industrial environment, compared with traditional wired networks, WHNCS has low reliable communication because of interferences such as noise, electromagnetic radiation, multipath distortion, temperature, and humidity from nearby industrial equipment and surrounding walls [47,102]. These situations result in packet loss and delay, making the adoption of WHNCS in the industrial environment as a challenging one. Signal parameters such as the link quality indicator (LQI) and radio signal strength indicator (RSSI) can be utilized to identify the link quality and reliable data transmission [103,104]. Other methods, such as path re-routing [39], efficient re-transmission techniques [105], link failure analysis, and redundancy devices [106], will be helpful to improve the reliability and data transmission. This might result in additional transmission overhead that wastes mote energy and makes the network congested, which in turn affects the reliable transmission of data [107]. In the future, the necessity of efficient routing algorithms in multi-hop communication with optimized memory utilization is expected to improve the processing power and reliability of motes.

### 4.3. Latency

Closed-loop processes continuously require real-time reliable sensor data to keep the process stable. If the network experiences any delay in data transmission larger than a specified time, the data cannot be used for effective control actions. Therefore, new critical data must be transmitted through the network to a sink instead of retrying all transmissions [37]. Additionally, packet delay and transmission failure result in the performance degradation of the WHNCS [48]. Forward error correcting, multi-path and multi-SPEED routing protocol (MMSPEED), and routing protocol for low-power and lossy networks (RPL), can be used to minimize the number of retransmission attempts to mitigate the network failure delay [39]. Another possible solution is designing a controller with a predictive nature to compensate the network and plant delay, which will minimize the impacts of unavoidable stochastic delay in the control loop [14,108].

### 4.4. Interoperability

Though industrial process plants contain multiple field devices which support different wireless communication protocols, all the IWSN protocols use the standard 2.4 GHz ISM frequency band as a physical layer. Recent significant improvements were made in the network layer, MAC layer, and physical layer to enhance these standards coexistence. These advancements are based on improving the routing methods, MAC layer restructuring, device sleep scheduling, and transmission power control. However, there is still a much wider gap that needs to be reduced to make all the IWSNs comply with each other, and carrying out more research on real-time scheduling for WHNCS is a high priority [109,110]. Interoperability will also reduce the necessity of procuring new devices for adopting new standards in the industrial environment [111].

### 4.5. Cost Effectiveness and Resource Utilization

The prime motivation for transitioning from wired to wireless solutions is the low cost requirements for deploying and installing wireless field devices. IWSN is intended to provide increased productivity, reduced maintenance, and decreased operating costs [112]. Some wireless sensor solutions cost less than $200 for deployment at the field level, while the same wired device installation costs can be doubled because of additional laying and wiring costs [113]. Most WirelessHART field devices are compact and small in size, which is an added advantage when encountering factory space problems, helping to make installing a large-scale network of nodes easier [114]. However, this compactness reduces the computational capabilities because of the limited memory and battery power supply. This situation also causes the WHNCS to suffer from a limited operational range in harsh industrial environments, which makes real-time data delivery challenging [115].

### 4.6. Power Consumption and Battery lifetime

One of the most critical parameters to be considered when adopting WHNCS is energy efficiency. Almost all wireless motes support battery operation capabilities with a low power consumption [116]. In order to conserve the energy of the motes, clustering algorithms such as energy-efficient sleep awake aware (EESAA), sleep-wake energy-efficient distributed (SEED), and hybrid energy-efficient distributed (HEED) can be adopted to minimize redundant transmission to achieve energy efficiency and improve data collection [117,118]. Adaptive free-shape clustering (AFC) is an emerging technique that greatly minimizes power consumption and increases the network lifetime [119]. Other non-conventional power generation methods such as photovoltaics, wind power, and thermoelectric conversion can be combined with WirelessHART field devices to achieve long-term operation without the need of any human intervention [82,120]. A detailed survey of energy harvesting in IWSN is discussed in [121]. Thus, power consumption and battery lifetime are a significant bottleneck while using the sensor nodes for their extended features at the field level.

### 4.7. Fault Tolerance

Network failures, mote power dropout, and stochastic delay in the WHNCS are unpredictable. Failure of one or more motes could result in the collapse of the entire process. Such characteristics have led to the development of various techniques, such as energy aware routing for low-energy ad hoc sensor networks (EAR-LEAHSN), the energy-aware QoS routing protocol (EQoSR), the multi-level route-aware clustering algorithm (MRLC), and the distributed clustering-based multipath algorithm (DCM), to provide standby redundant paths/devices to support multi-hopping [122,123]. Using these techniques in real time comes with complications, such as increased energy consumption due to the multiple copies of data transmission to sink node, greater bandwidth utilization for reliable data, continuous alternate path-finding, and complex data reconstruction processes [124].

### 4.8. Data Accessibility

Data accessibility and management are some of the most frequently occurring issues in the WHNCS because of the limited storage availability and may even lead to end-to-end packet delay [81]. Sensors installed in particular industrial environments can send identical data, which often leads to unnecessary data aggregation and the need to process a massive amount of metadata. To avoid this problem, motes can be designed so as to filter the non-critical data using predetermined conditions or distributed source coding (DSC) to compress the raw data before sending them to the sink [125]. These methods can significantly improve data scalability, device versatility, and battery life [126]. Alternative techniques, such as low-energy adaptive clustering hierarchical (LEACH), power efficient gathering in sensor information systems (PEGASIS), and multi-hop routing protocol with unequal clustering (MRPUC), can be combined into clusters in order to overcome the problem of data transmission for a large group of nodes [127]. However, the research trends show that an increase in the network size increases the computational and communication overhead of the WHNCS.

### 4.9. Autonomous and Predictive Characteristics

Unexpected network/mote collapse results in catastrophic failures in the closed-loop process, which creates the need for the independent operation of motes as autonomous and self-organizing devices without any human intervention [128]. In this situation, the addition or removal of the deployed motes may lead to network partitioning. WirelessHART has the advantage of mesh networking, which enables it to form a self-organizing network coordination framework (SoNCF). The framework can independently create multiple packet time slots in a self-organized manner for better data prediction throughput [129]. The implementation of autonomous nodes comes with greater energy consumption and more extensive data aggregation due to their continuous monitoring and transmission. Most WHNCS challenges and design goals are interlinked with one another, causing them to evolve continuously with new technological advancements to acquire the same promising performance as the wired networks.

## 5. Summary and Conclusions

This survey presents the application of WirelessHART from the perspective of industrial monitoring and control in both simulations and real-time environments. This paper also examines the disparity between industrial needs and the currently available technology, which creates challenges for the performance of WHNCS. The design goals and application challenges faced by WirelessHART were comprehensively addressed. Additionally, possible research developmental solutions, which may solve most of the above-stated problems, are given in Table 8 to improve its performance and increase the chance of adoption by the process control industries. Additionally, though WirelessHART is the leader in its field, IWSN is still evolving. It is too early to suggest which wireless protocol will most impact the process control industry in the future. More factors, such as interoperability between the various standards, scalability, security, reliability, and real-time updates with the limited latency range prescribed by industrial requirements, were also kept in mind while enhancing the standards. To overcome the above challenges and requirements, an efficient WirelessHART network can be introduced to all process industries and will hopefully eventually replace the existing wired legacy systems. However, the current WHNCS development rate is too slow to reach its maximum potential and needs continuous improvements. It also requires collaborative progress with other IWSN protocols to be widely accepted in the industrial process control.

## Figures and Tables

**Figure 1 sensors-21-04951-f001:**
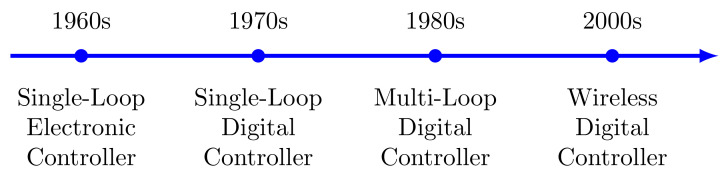
Evolution of wireless digital controllers.

**Figure 2 sensors-21-04951-f002:**
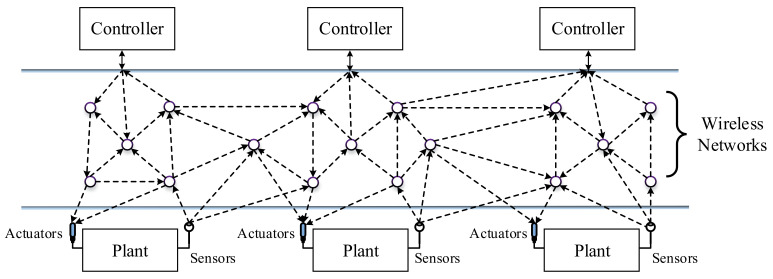
IWSN architecture.

**Figure 3 sensors-21-04951-f003:**
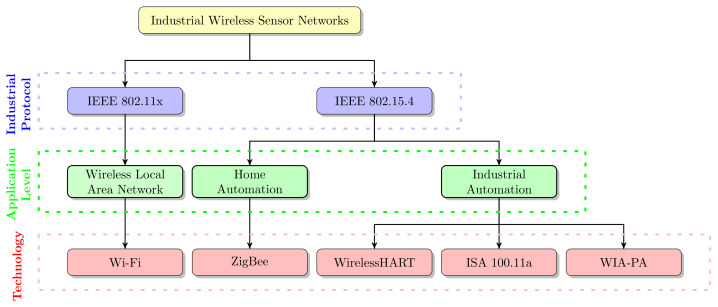
Classification of industrial wireless sensor networks.

**Figure 4 sensors-21-04951-f004:**
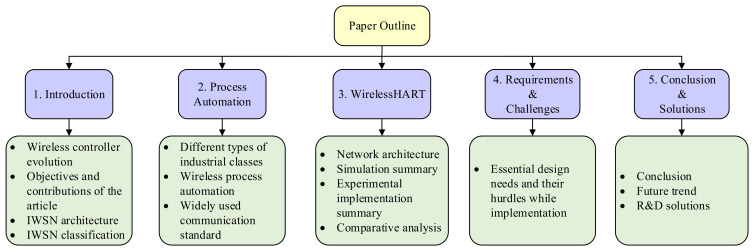
Content organization of the paper.

**Figure 5 sensors-21-04951-f005:**
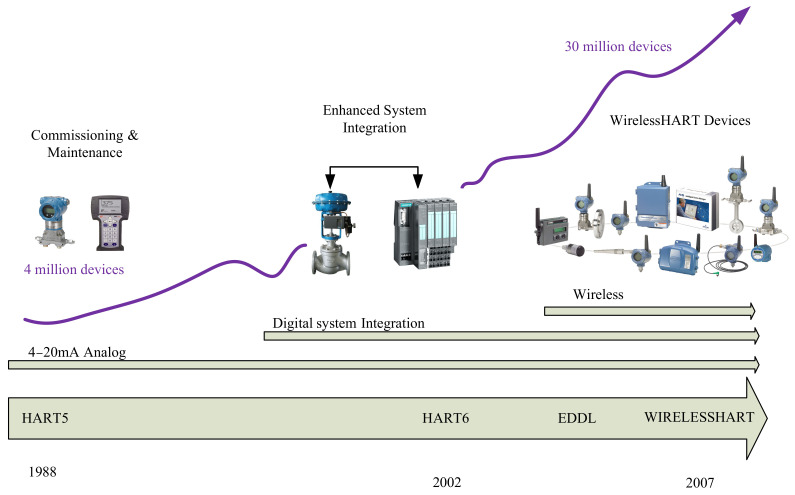
Evolution of the HART protocol [14].

**Figure 6 sensors-21-04951-f006:**
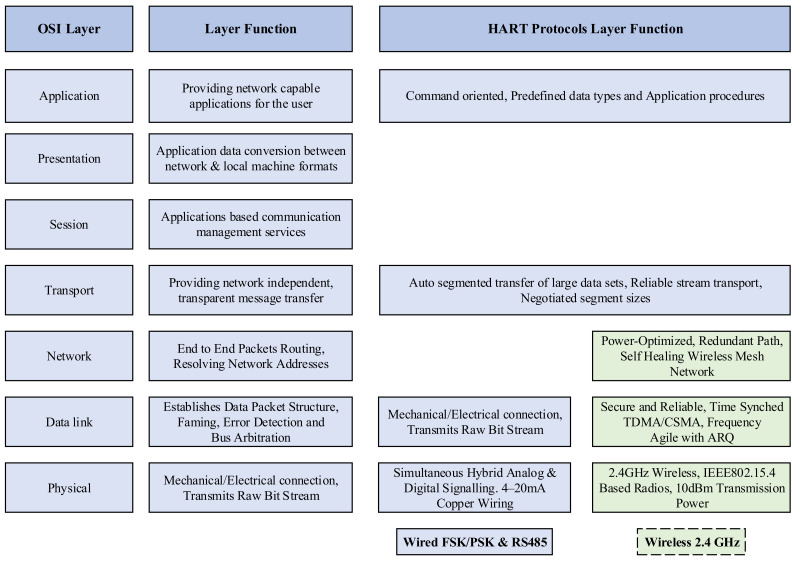
OSI layer of the conventional wired HART and the WirelessHART protocols [43].

**Figure 7 sensors-21-04951-f007:**
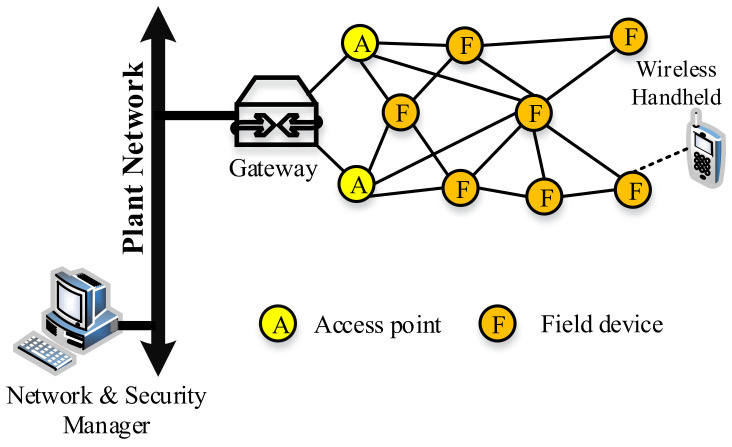
WirelessHART network structure.

**Figure 8 sensors-21-04951-f008:**
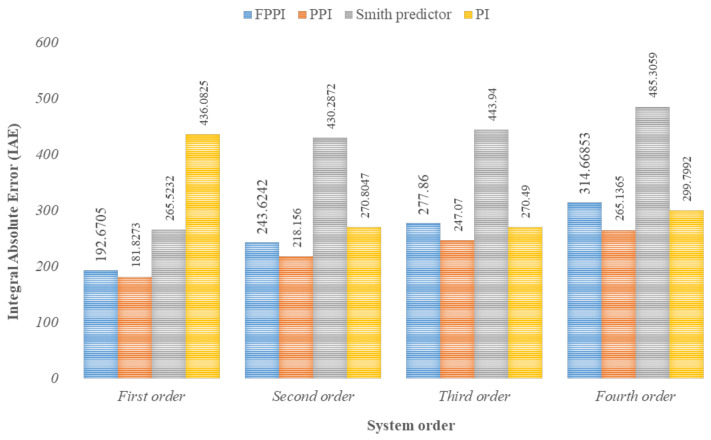
IAE values of various controllers in WirelessHART network [14].

**Table 1 sensors-21-04951-t001:** Adopting factors for wireless technology in industrial automation [13].

Factors	2012 (%)	2014 (%)
Data Accuracy	96	96
Data Protection	89	87
Data Accessibility	71	69
Industrial Standards	68	62
Cost Effective	61	59
IP Compatibility	45	54
Battery Lifetime	68	51

**Table 2 sensors-21-04951-t002:** Features comparison between the different industrial wireless sensor networks [14].

Features	Standard			
	ZigBee	WirelessHART	ISA100.11a	WIA-PA
Data Security	High	Very High	Very High	Very High
Scalability	Medium	High	High	High
Power Usage	Low	Low	Low	Low
Data Transfer Rate	Low (20–250 kbps)			
Network Topology	Star/Mesh/Tree	Star/Mesh	Star/Mesh	Hybrid
Data Reliability	Low	Very High	Very High	Very High
Routing Capability	Limited	Full	Full/Limited	Limited
Channel Hopping	No	Yes	Yes	Yes
Frequency channels	27 (All Bands)	15 (2.4 GHz)	16 (2.4 GHz)	16 (2.4 GHz)
Manager Architecture	Centralized/ Distributed	Centralized	Centralized	Centralized/ Distributed

**Table 3 sensors-21-04951-t003:** Classes of industrial process automation.

Category	Latency	Class	Description	End Function	Field Devices
Safety and Supervisory	10 ms	Emergency control	Always critical	Emergency shutdown	Vibration sensor Gas sensor Sprinklers
Closed-loop Control	10–100 ms	Regulatory control	Often critical	Field device control	Control valve Flow meter
		Supervisory control	Mostly non-critical	Control loops optimization	
Monitoring and Control	100–1000 ms	Open-loop control	Corrective maintenance	Manual process shutdown	Proximity sensor DC motor Relays
		Alerting systems	Preventive maintenance	Regular maintenance, Field device examinations		
		Monitoring systems	Periodic maintenance	Record maintenance, Event sequence recording	

**Table 4 sensors-21-04951-t004:** Preferred IWSN standards [12].

Wireless Standard	2012 (%)	2014 (%)
WirelessHART	27	25
ISA 100.11a	10	11
Hybrid	22	16
Others (WIA-PA and ZigBee)	23	28
Factory Automation	13	13

**Table 5 sensors-21-04951-t005:** Summary of WirelessHART for monitoring and control applications in the simulation environment.

Ref.	Process	Field device	T	ST	C	Mo	Challenges Addressed	Results
[50]	Network analysis	Temperature and Pressure Sensors	M	Network simulator (NS-1)	-	SI	Security (Signal Jamming), Interoperability	WirelessHART and WLAN coexistence investigation and network performance examination
[51]		Emerson Smart Wireless Gateway	S	Emerson (AMS Snap-on)	-	RT	Packet loss	Data scheduling and routing analysis for packet drop mitigation
[52]		Sensor nodes	M	Network simulator (NS-2)	-	SI	Latency, Noise	Effects of packet error rate and packet drop analysis
[53]	Sensor node networking experiment	Temperature sensor, XDM2510HE Dust network gateway	L, M, S	WirelessHART network simulator	-	SI, RT	Signal reliability, Latency	WirelessHART network for Dense Reader Environment in industrial monitoring
[56]	Laboratory scale open loop process	ABB AC800M	M	TrueTime with MATLAB	PI, PPI	SI	Latency	Reduced the problems caused by clock drift
[55]	DC Motor control	Sensor nodes	M	TrueTime with MATLAB	PD	SI	Packet loss, Channel hopping	WirelessHART implementation for sluggish processes
[58]	Level Process	Sensor nodes	M	Jitterbug toolbox	LQG	SI	Data reliability	Improving the network reliability and controller design

T, topology; L, linear; M, mesh; S, star; C, controller; Mo, model of research; ST, simulation tool; SI, simulation; RT, real time.

**Table 6 sensors-21-04951-t006:** Tunable parameters of various controllers [70].

Controller	Model Parameters			Controller Parameters		
PI	-	-	-	$K_{c}$	$T_{i}$	-
FOPI	-	-	-	$K_{c}$	$T_{i}$	λ
PPI	-	-	$L_{p}$	$K_{c}$	$T_{i}$	-
FPPI	-	$T_{f}$	$L_{p}$	$K_{c}$	$T_{i}$	
FOPPI	-	-	-	$K_{c}$	$T_{i}$	λ
Smith predictor	*K*	*T*	$L_{p}$	$K_{c}$	$T_{i}$	-
IMC	*K*	*T*	$L_{p}$	-	$T_{cl}$	-

**Table 7 sensors-21-04951-t007:** Summary of WirelessHART for monitoring and control applications.

Ref.	Process	Field Device	T	ST	C	Mo	Challenges Addressed	Results
[59]	Actuator to sensor communication	Sensor nodes	-	WirelessHART simulator in MATLAB	-	SI	Reliability (Interference minimization)	WirelessHART simulator for large scale networks
[61]	DC servo motor control	Virtual sensor nodes	M	TrueTime with OMNET++ in MATLAB	PI	SI	Interoperability	Improving coexistence management for WirelessHART and ISA100.11a
[62]	Unstable batch reactor	Sensor nodes	L	Network simulator	PI	SI	Stability	Stabilized routing configuration using a hybrid approach for non-linear systems and time-varying transmissions
[11]	Network analysis on industrial process models	XDM2510H WirelessHART RF Module	-	WirelessHART simulator in MATLAB	PID	SI	Link reliability, Latency	Effective design of an EWMA filter to mitigate packet dropout and link delay
[63]	Test bed of TelosB mote	Chipcon CC2420	M	TinyOS 2.2, TOSSIM	-	SI	Link reliability, Channel Hopping	Time window allocation for real-time scheduling to reduce resource usage and enhancing the scalability
[64]	Laboratory scale testing	Freescale MC1321x evaluation toolkit	M	ANSI C in HCS08	-	RT	Security	Data scheduling and transmission between the field devices to indicate the possibility of monitoring and data transfer
[65]	WirelessHART test bench	Rosemount 648 TT	M	Wi-Analys	-	RT	-	Achieving data transfer between the field nodes and the controller via gateway
[66]	Steam flow and pressure of Distillation column	Raschig Jaeger RSP-250	-		PID	RT	Latency analysis, Data reliability	To control the industrial process in WirelessHART network
[67]	DC servo motor	NI-DAC control cards	-	LabVIEW	P	RT	Delay, Packet loss	Performance investigation for the effect of different networks on the control with a simulation study for fixed packet loss case
[68]	Industrial process plant	-	-	DeltaV control system	PID, PIDPlus	RT	Data reliability	Designing a control strategy for non-periodic measurement updates from the processes
[14]	Pressure process plant	Linear tech Smart Mesh WirelessHART (XG2510HE gateway and XDM2510H node)	M	MATLAB	PI, PPI, FPPI	SI, RT	Predictive characteristics, Packet delay	Controlling a real-time process plant even in presence of noise and packet delay
[69]	Distillation column control	Wireless field devices	-	DeltaV PredictPro	PID, MPC	RT	Predictive characteristics	To control the industrial process with real-time data prediction for packet loss compensation
[71]	Network simulation	Sensor nodes	-	TrueTime with MATLAB	PID	SI	Packet drop	Simulation study on wired and wireless networked control system under various packet loss conditions
[72]		WirelessHART mote (DC9003A) Eterna Interface card (DC9006A)	M	SmartMesh API Explorer stack with MATLAB	PID	SI	Network delay	Network induced delays measurement technique and its effects on a pilot process plant
[48]	Level Flow, Heat and pressure process plants	Linear Technology WirelessHART Modules	M	MATLAB	PID, MPC	SI, RT	Latency, Data reliability	Even under model mismatch and packet delay variation, the controller is designed to keep the process control loop as a stable one
[73]	Network analysis	AwiaTech WirelessHART evaluation kit	L, M, S	AwiaTech WirelessHART simulator	-	SI, RT	Latency, Data reliability, Interference	Examined the joining time for each node and their effect on the distance between them
[37]	Industrial processes transfer functions	Linear tech Smart Mesh WirelessHART (XG2510HE gateway, XDM2510H node)	-	WirelessHART hardware-in- the-loop simulator with MATLAB	PI, Smith predictor, Set-point weighted PI	SI, RT	Stochastic delay, Noise	Controller implementation in WirelessHART network under model mismatch, stochastic delay, and noise conditions
[74]	Flow process	SmartMesh WirelessHART kit	-	SmartMesh API Explorer stack with MATLAB	PI, PID, Fuzzy PID	RT	Network delay	Investigations on the effects of using wired and WirelessHART motes on the control performance on pilot process plant
[75]	Tennessee Eastman (TE) Plant	Sensor nodes	M	OMNET++ wireless network simulator	-	SI	Packet errors, Packet drop, Link failure	Performance study of WirelessHART networks on a TE plant in the presence of packet errors and packet drop
[76]	Flow and Level process	WirelessTHUM adaptor	-	LabVIEW and Emerson Smart Wireless Gateway	PID	RT	Network delay	Cascaded PID controller is designed and experimented to handle network induced delay and disturbance
[77]	Teleoperated system	-	-	MATLAB and SystemC	-	SI	Packet loss, Delay	Networked control systems co-simulations for time synchronization and error tracking
[78]	Batch reactor	-	M	WirelessHART simulator	LQR	SI	Packet scheduling and transmission	Scheduling and routing in a WirelessHART networked control system with controller co-design
[79]	Valve actuation control	Emerson 1420A	M	HART UDP interface with C++	PID	RT	Latency	Investigation of control valve positioning using a PID controller in a WirelessHART environment
[80]	Wireless sensor network testbed	TelosB motes with Chipcon CC2420	M	WirelessHART network simulator	-	SI	Latency, Interference	Determining the packet schedulability of real-time data flow based on new network model map
[81]	Network analysis	Virtual sensor nodes	Random	MATLAB	-	SI	Data scheduling, Latency	Examination of the problem in joint transmission scheduling and channel allocation to minimize end-to-end delay
[82]			S, M, T	NS-2 simulator and OPNET simulator	-	SI	Delay, Energy consumption	Performance comparison of different industrial wireless sensor network protocols
[83]		Rosemount —1420 Rosemount —3051S	S, M	DeltaV and Emerson Smart Wireless Gateway	PID	SI, RT	Latency, Packet drop, Noise	New controller design to overcome network delay and packet dropout even in a noisy environment
[84]	Industrial process transfer functions	WirelessHART mote (DC9003A) Eterna Interface card (DC9006A)	M	SmartMesh API Explorer with MATLAB	Fuzzy Set-point weighted PI	SI	Latency, Packet drop	Mitigation of network delay and packet drop in the closed-loop process using a Fuzzy Adaptive Set-point Weighting Controller
[85]	Flow process	SmartMesh WirelessHART kit, T-click board with Arduino	-	MATLAB	PI, Smith predictor, Fuzzy PID	RT	Packet delay, Latency	Implementation of locally developed WirelessHART adaptors in the pilot process plant and performance comparison of various controllers
[86]	Industrial process transfer functions	Virtual sensor nodes	M	Python	PID	SI	Network load, Link reliability, Latency	Examination of variable payload message length effects in round trip delay measurements
[87]	Transfer function of thermal chamber process	Linear tech Smart Mesh WirelessHART kit (XG2510HE gateway, XDM2510H node)	-	MATLAB	Smith predictor, PI, PPI	SI	Delay, Noise, Disturbance	New controller performance analysis over a variable network delay, external noise, and process dead-time
[88]	Valve control	Rosemount 702, Fisher 4320 on/off valve, AwiaTech WirelessHART Evaluation Kit	S	Emerson Process Management	P	RT	Latency, Data Reliability, Signal Interference	Comparative study of WirelessHART and wired Foundation Fieldbus for valve control
[89]	Flow process	Fisher 4320 wireless position transmitter	-	DeltaV control system	PID, PIDPlus	RT	Delay	Experimentation of valve position control using a PIDPlus controller in a WirelessHART network for the flow process
[90]	Production decision and supporting system (PDSS)	34 TelosB motes with Chipcon CC2410	M	TinyOS 2.1 with CC2420x radio driver	LQR	RT	Latency, Data transmission	Co-design strategies for a small industrial Cyber–Physical System to enhance communication reliability
[91]	Two tank system	Emerson Smart Wireless Gateway kit	-	LabVIEW	-	RT	Delay	Modeling and flow measurement of a coupled tank process based on Laplace transformation with simple linear optimizations to reduce sudden load disturbance and errors

**Table 8 sensors-21-04951-t008:** WirelessHART challenges and possible research and development solutions.

Challenges	Limitations/ Problems	Possible Research and Development Solutions
Battery	Limited power supply Price is proportional to capacity and durability	Sleep scheduling [63,78] Passive data transfer [130,131] Effective data redundancy [78]
Memory	Limited memory power for complex processes	Memory optimization and additional memory allocation [63,132,133]
Computational power	Confined traditional processor	Usage of modern SRAM and DRAM [134] Current generation processor co-design [58,78,90]
Data transmission	Interference Overlapping	Adaptive channeling and multi-hop communication [135,136] Distribute routing protocol [39,137]
Delay	Process instability Stochastic delay	Slotted retransmission and scheduled transmissions [138] Priority data access [139] Fault tolerant [53,127]
Network traffic	Random transmission Interference and overlapping Data aggregation	Channel scheduling and TDMA slotting [51,81] Estimation and filtering [37,135] (e.g., Kalman, Particle)
Controlling	Delay Network/ mote failure	Multi-hop transmission [135,140] Delay compensators [85,87] Model-based predictive controllers [14,48]
Security	DoS, QoS Data theft Channel flooding Hacking Signal interference	Data and Network encryption and authentication [71,141,142] Cryptographic keying [100,143]
Interoperability	Inadequate standardization Existing numerous protocols	IPv6-based enhancement [47,144] Interoperable node and network development [50,61,145]

## Data Availability

Not applicable.

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
