# Peer review of "A Survey on the Application of WirelessHART for Industrial Process Monitoring and Control"

_sensors, 2021, doi:10.3390/s21154951_

Round 1
Reviewer 1 Report
The review presents a survey that evaluates the application of WirelessHART in industrial process monitoring and control. The authors have done a good job, as is well demonstrated by the bibliography presented and the different summary tables. As well as table 10 with the limitations and problems and their possible research solutions.
On the other hand, the English in my opinion is poor and needs to be reviewed by a native.
The following is an example where the sentence is unconnected (lines 82-84):
A typical wireless network requires small to midscale infrastructure consists of multiple sensor nodes working together to acquire data from the field devices installed in different environments.
It is necessary to check some small errors:
Lines 35 and 36: The text says, “Some of the advantages of the IWSNs over the wired network’s are,”, but the papers indicated, [7-10], seem to be more focused on WirelessHART than on IWSN in general.
Line 89: “… and food processing [22], environment monitoring [23].” It is necessary to write “and” after the last comma.
Line 150: The references should be reorganized. “... can be found in [3], [6], [14], [32], and [33].”
Line 283: smith predictor... Smith should be capitalized.
It is confusing that Table 4, which should present information on WirelessHART simulation has the same title as Tables 6 to 9. The latter tables should show contributions focused on wirelessHART implementation, but the title is the same.
The reference [26] seems incomplete.
Author Response
Please find the attached response to the reviewer comments, along with the highlighted changes in the updated article.

Reviewer 2 Report
This paper claims to survey the application of wirelessHart for industrial process monitoring and control. Consequently, the paper is structured as is expected from a survey paper. The introduction is followed by a rather high-level background section on Process Automation. Section 3 contains what is supposed to be the core part of the paper: it introduces WirelessHART (but lacks any details) and it captures applications of WirelessHART in simulation as well as in real-time industrial environments. Section 4 presents further research directions before section 5 concludes.
Overall, the paper addresses an important question, namely how wireless networks can help to make IIoT a reality. The authors present a remarkable number of references to pursue their work. Yet, the paper has a significant number of weaknesses that disqualify its publication.
- The language of the paper is hard to grasp. This is not so much about vocabulary, but rather about the way of writing. The authors use long and very long sentences leading to paragraphs that lack an inner structure.
- The paper lacks a clear goal / motivation. After having read the abstract and the introduction it is still unclear what goal the authors pursue, neither do they present their targeted contributions. Unfortunately, the reader is left in that state until the end of the paper. Particularly, section 3 provides an extensive list of reference, but it is (i) unclear for what purpose, (ii) how the references relate to each other (are there works building on each other or is this just a mass of non-related independent research artefacts?), (iii) which benefit they provide (why should I as a researcher read this paper?).
- The paper lacks a description of the methodology used by the authors. How did you find all the papers used as references?
- The standard the authors are discussing has been released in 2007 (hence, it is really old). In case there have not been any updates to the standard since then, I am questioning the relevance of the protocol for the discussed usage scenarios and hence of the research based on the protocol. If there has been an update to the standard, I would expect the authors to discuss this and also to clarify which parts of the works they references has already been taken into account in the updates.
- Similar to the standard itself, many references are almost a decade old. Particularly, core references [6,13], the authors build their line of argumentation on have been released in 2012 and 2014.
- Others
- Figure 3 is not discussed
- I think the caption of Figure 4 is wrong.
- It remains unclear what the authors mean by “real-time” system
Major suggestions to the authors for a revised version of the paper besides fixing the shortcomings mentioned earlier:
- Barely any background is presented on WirelessHART. The discussion in 3.1 is very shallow. I would appreciate more depth here as well as a description how to map WirelessHART to the layers of the OSI/ISO model.
- please introduce more structure to your survey. You may consider going as far as introducing (or adopting) a taxonomy. Further, please group references with similar content and approaches. There is no need to write one or more sentences to all references. IMHO, this does not help the reader getting up to speed with WirelessHART.
Author Response

(The authors gave the same response as above.)

Reviewer 3 Report
The authors present a detailed study focusing only on WirelessHART adoption in simulation and real-time applications for industrial process monitoring and control with its crucial challenges and requirements.
Significant: Yes, the paper is a significant review.
Supported: Mostly yes,
Referencing: some additions are necessary
Quality: The organization of the manuscript and presentation some improvement.
I suggest to the authors to increase the sections 3 and 4.
I suggest the authors the following papers for reading:
Industrial Wireless Sensor Networks: Protocols and Applications. Sensors 2020, 20, 5809. https://doi.org/10.3390/s20205809
Survey on Wireless Technology Trade-Offs for the Industrial Internet of Things. Sensors 2020, 20, 488. https://doi.org/10.3390/s20020488
Evaluation of Suitability of Current Industrial Standards in Designing Control Applications for Internet of Things Healthcare Sensor Networks. J. Sens. Actuator Netw. 2019, 8, 54. https://doi.org/10.3390/jsan8040054
Sensors Network for Monitoring the Carasau Bread Manufacturing Process. Electronics 2019, 8, 1541. https://doi.org/10.3390/electronics8121541
Microclimate Monitoring and Modeling through an Open-Source Distributed Network of Wireless Low-Cost Sensors and Numerical Simulations. Eng. Proc. 2020, 2, 18. https://doi.org/10.3390/ecsa-7-08270
Supervisory Control for Wireless Networked Power Converters in Residential Applications. Energies 2019, 12, 1911. https://doi.org/10.3390/en12101911
A Polling-Based Transmission Scheme Using a Network Traffic Uniformity Metric for Industrial IoT Applications. Sensors 2019, 19, 187. https://doi.org/10.3390/s19010187
A Novel Charging Method for Underwater Batteryless Sensor Node Networks. Sensors 2021, 21, 557. https://doi.org/10.3390/s21020557
WSN Hardware for Automotive Applications: Preliminary Results for the Case of Public Transportation. Electronics 2019, 8, 1483. https://doi.org/10.3390/electronics8121483
A New Method of Priority Assignment for Real-Time Flows in the WirelessHART Network by the TDMA Protocol. Sensors 2018, 18, 4242. https://doi.org/10.3390/s18124242
Author Response

(The authors gave the same response as above.)

Round 2
Reviewer 2 Report
I think the updates provided by the authors are sufficient to accept the paper.
Author Response

(The authors gave the same response as above.)

Reviewer 3 Report
The paper has been improved, but I suggest the authors increase the introduction , and check the references of the first revision. The quality of the work has improved a lot.
Author Response

(The authors gave the same response as above.)
